# GeCoNeRF: Few-Shot Neural Radiance Fields via Geometric Consistency

## Abstract

We present a novel framework to regularize Neural Radiance Field (NeRF) in a few-shot setting with a geometry-aware consistency regularization. The proposed approach leverages a rendered depth map at unobserved viewpoint to warp sparse input images to the unobserved viewpoint and impose them as pseudo ground truths to facilitate learning of NeRF. By encouraging such geometry-aware consistency at a feature-level instead of using pixel-level reconstruction loss, we regularize the NeRF at semantic and structural levels while allowing for modeling view-dependent radiance to account for color variations across viewpoints. We also propose an effective method to filter out erroneous warped solutions, along with training strategies to stabilize training during optimization. We show that our model achieves competitive results compared to state-of-the-art few-shot NeRF models.

## 1 Introduction

Recently, representing a 3D scene as a Neural Radiance Field (NeRF) Mildenhall et al. (2020) has proven to be a powerful approach for novel view synthesis and 3D reconstruction Barron et al. (2021); Jain et al. (2021); Chen et al. (2021). However, despite its impressive performance, NeRF requires a large number of densely, well distributed calibrated images for optimization, which limits its applicability. When limited to sparse observations, NeRF easily overfits to the input view images and is unable to reconstruct correct geometry Zhang et al. (2020).

The task that directly addresses this problem, also called a few-shot NeRF, aims to optimize high-fidelity neural radiance field in such sparse scenarios Jain et al. (2021); Kim et al. (2022); Niemeyer et al. (2022), countering the underconstrained nature of said problem by introducing additional priors. Specifically, previous works attempted to solve this by utilizing a semantic feature Jain et al. (2021), entropy minimization Kim et al. (2022), SfM depth priors Deng et al. (2022) or normalizing flow Niemeyer et al. (2022), but their necessity for handcrafted methods or inability to extract local and fine structures limited their performance.

To alleviate these issues, we propose a novel regularization technique that enforces a geometric consistency across different views with a depth-guided warping and a geometry-aware consistency modeling. Based on these, we propose a novel framework, called Neural Radiance Fields with Geometric Consistency (GeCoNeRF), for training neural radiance fields in a few-shot setting. Our key insight is that we can leverage a depth rendered by NeRF to warp sparse input images to novel viewpoints, and use them as pseudo ground truths to facilitate learning of fine details and high-frequency features by NeRF. By encouraging images rendered at novel views to model warped images with a consistency loss, we can successfully constrain both geometry and appearance to boost fidelity of neural radiance fields even in highly under-constrained few-shot setting. Taking into consideration non-Lambertian nature of given datasets, we propose a feature-level regularization loss that captures contextual and structural information while allowing for modeling view-dependent color differences. We also present a method to generate a consistency mask to prevent inconsistently warped information from harming the network. Finally, we provide coarse-to-fine training strategies for sampling and pose generation to stabilize optimization of the model.

We demonstrate the effectiveness of our method on synthetic and real datasets Mildenhall et al. (2020); Jensen et al. (2014). Experimental results prove the effectiveness of the proposed model over the latest methods for few-shot novel view synthesis.

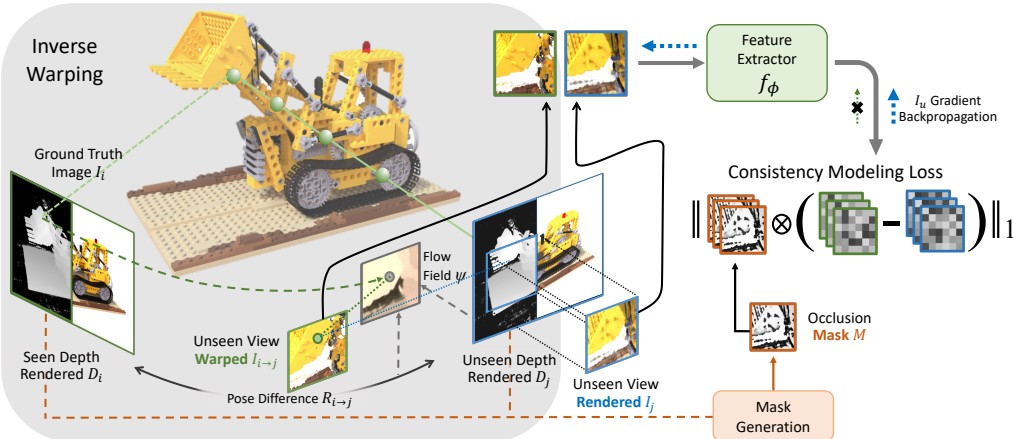

Figure 1: **Illustration of our consistency modeling pipeline for few-shot NeRF.** Given an image $I_i$ and estimated depth map $D_j$ of $j$-th unobserved viewpoint, we warp the image $I_i$ to that novel viewpoint as $I_{i \to j}$ by establishing geometric correspondence between two viewpoints. Using the warped image as a pseudo ground truth, we cause rendered image of unseen viewpoint, $I_j$, to be consistent in structure with warped image, with occlusions taken into consideration.

## 2  RELATED WORK

**Neural radiance fields.**    Among the most notable of approaches regarding the task of novel view synthesis and 3D reconstruction is Neural Radiance Field (NeRF) Mildenhall et al. (2020), where photo-realistic images are rendered by a simple MLP architecture. Sparked by its impressive performance, a variety of follow-up studies based on its continuous neural volumetric representation have been prompted, including dynamic and deformable scenes Park et al. (2021); Tretschk et al. (2021); Pumarola et al. (2021); Attal et al. (2021), real-time rendering Yu et al. (2021a); Hedman et al. (2021); Reiser et al. (2021); Müller et al. (2022), self-calibration Jeong et al. (2021) and generative modeling Schwarz et al. (2020); Niemeyer & Geiger (2021); Xu et al. (2021); Deng et al. (2021). Mip-NeRF Barron et al. (2021) eliminates aliasing artifacts by adopting cone tracing with a single multi-scale MLP. In general, most of these works have difficulty in optimizing a single scene with a few number of images.

**Few-shot NeRF.**    One key limitation of NeRF is its necessity for large number of calibrated views in optimizing neural radiance fields. Some recent works attempted to address this in the case where only few observed views of the scene are available. PixelNeRFYu et al. (2021b) conditions a NeRF on image inputs using local CNN features. This conditional model allows the network to learn scene priors across multiple scenes. Stereo radiance fields Chibane et al. (2021) use local CNN features from input views for scene geometry reasoning and MVSNeRF Chen et al. (2021) combines cost volume with neural radiance field for improved performance. However, pre-training with multi-view images of numerous scenes are essential for these methods for them to learn reconstruction priors.

Other works attempt different approach of optimizing NeRF from scratch in few-shot settings: DSNeRF Deng et al. (2022) makes use of depth supervision to network to optimize a scene with few images. Roessle et al. (2021) also utilizes sparse depth prior by extending into dense depth map by depth completion module to guide network optimization. On the other hand, there are models that tackle depth prior-free few-shot optimization: DietNeRF Jain et al. (2021) enforces semantic consistency between rendered images from unseen view and seen images. RegNeRF Niemeyer et al. (2022) regularizes the geometry and appearance of patches rendered from unobserved viewpoints. InfoNeRF Kim et al. (2022) constrains the density's entropy in each ray and ensures consistency across rays in the neighborhood. While these methods constrain NeRF into learning more realistic geometry, their regularizations are limited in that they require extensive dataset-specific fine-tuning and that they only provide regularization at a global level in a generalized manner. Improving upon above works, our method tackles prior-free few-shot optimization without using any depth priors, achieving more local and scene-specific regularization with warping-based consistency modeling.

**Self-supervised photometric consistency.**    In the field of multiview stereo depth estimation, consistency modeling between stereo images and their warped images has been widely used for self-

supervised training Godard et al. (2017); Garg et al. (2016); Zhou et al. (2017) In weakly supervised or unsupervised settings Huang et al. (2021); Khot et al. (2019) where there is lack of ground truth depth information, consistency modeling between images with geometry-based warping is used as a supervisory signal Zhou et al. (2017); Huang et al. (2021); Khot et al. (2019) formulating depth learning as a form of reconstruction task between viewpoints.

Recently, methods utilizing self-supervised photometric consistency have been introduced to NeRF: concurrent works such as NeuralWarp Darmon et al. (2022), StructNeRF Chen et al. (2022) and Geo-NeuS Fu et al. (2022) model photometric consistency between source images and their warped counterparts from other source viewpoints to improve their reconstruction quality. However, these methods only discuss dense view input scenarios where pose differences between source viewpoints are small, and do not address their behavior in few-shot settings - where sharp performance drop is expected due to scarcity of input viewpoints and increased difficulty in the warping procedure owing to large viewpoint differences and heavy self-occlusions. RapNeRF Zhang et al. (2022) uses geometry-based reprojection method to enhance view extrapolation performance, and Bortolon et al. (2022) uses depth rendered by NeRF as correspondence information for view-morphing module to synthesize images between input viewpoints. However, these methods do not take occlusions into account, and their pixel-level photometric consistency modeling comes with downside of suppressing view-dependent specular effects.

## 3 PRELIMINARIES

Neural Radiance Field (NeRF) Mildenhall et al. (2020) represents a scene as a continuous function $f_\theta$ represented by a neural network with parameters $\theta$, where the points are sampled along rays, represented by $r$, for evaluation by the neural network. Typically, the sampled coordinates $\mathbf{x} \in \mathbb{R}^3$ and view direction $\mathbf{d} \in \mathbb{R}^2$ are transformed by a positional encoding $\gamma$ into Fourier features Tancik et al. (2020) that facilitates learning of high-frequency details. The neural network $f_\theta$ takes as input the transformed coordinate $\gamma(\mathbf{x})$ and viewing directions $\gamma(\mathbf{d})$, and outputs a view-invariant density value $\sigma \in \mathbb{R}$ and a view-dependent color value $\mathbf{c} \in \mathbb{R}^3$ such that

$$\{\mathbf{c}, \sigma\} = f_\theta(\gamma(\mathbf{x}), \gamma(\mathbf{d})). \tag{1}$$

With a ray parametrized as $\mathbf{r}_p(t) = \mathbf{o} + t\mathbf{d}_p$ from the camera center o through the pixel $p$ along direction $\mathbf{d}_p$, the color is rendered as follows:

$$C(\mathbf{r}_p) = \int_{t_n}^{t_f} T(t)\sigma(\mathbf{r}_p(t))\mathbf{c}(\mathbf{r}_p(t), \mathbf{d}_p)dt,$$

$$\text{where } T(t) = \exp\left(-\int_{t_n}^{t} \sigma(\mathbf{r}_p(s))ds\right), \tag{2}$$

where $C(\mathbf{r}_p)$ is a predicted color value at the pixel $p$ along the ray $\mathbf{r}_p(t)$ from $t_n$ to $t_f$, and $T(t)$ denotes an accumulated transmittance along the ray from $t_n$ to $t$. To optimize the networks $f_\theta$, the observation loss $\mathcal{L}_{\text{obs}}$ enforces the rendered color values to be consistent with ground truth color value $C'(\mathbf{r})$:

$$\mathcal{L}_{\text{obs}} = \sum_{\mathbf{r}_p \in \mathcal{R}} \|C'(\mathbf{r}_p) - C(\mathbf{r}_p)\|_2^2, \tag{3}$$

where $\mathcal{R}$ represents a batch of training rays.

## 4 METHODOLOGY

### 4.1 MOTIVATION AND OVERVIEW

Let us denote an image at $i$-th viewpoint as $I_i$. In a few-shot novel view synthesis, NeRF is given only a few images $\{I_i\}$ for $i \in \{1, ..., N\}$ with small $N$, e.g., $N = 3$ or $N = 5$. The objective of novel view synthesis is to train the mapping function $f_\theta$ that can be used to recover an image $I_j$ at $j$-th unseen or novel viewpoint. As we described above, in the few-shot setting, given $\{I_i\}$, directly optimizing $f_\theta$ solely with the pixel-wise reconstruction loss $\mathcal{L}_{\text{obs}}$ is limited by its inability to model view-dependent effects, and thus an additional regularization to encourage the network $f_\theta$ to generate consistent appearance and geometry is required.

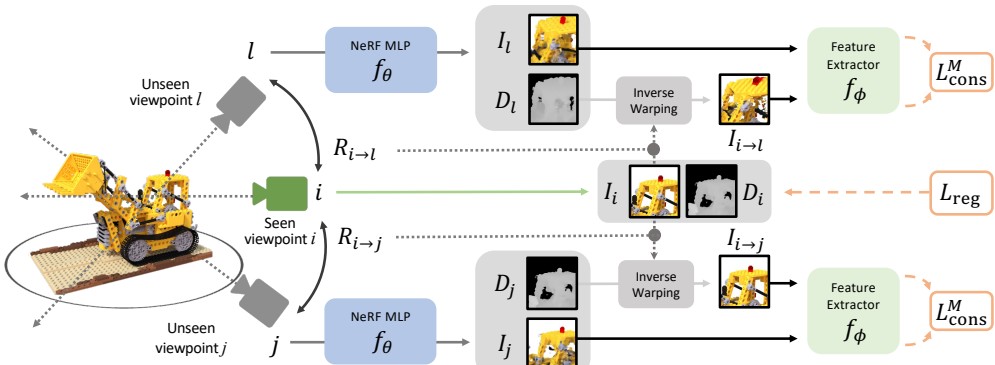

Figure 2: **Illustration of the proposed framework.** GeCoNeRF regularizes the networks with consistency modeling. Consistency loss function $\mathcal{L}_{\text{cons}}^M$ is applied between unobserved viewpoint image and warped observed viewpoint image, while disparity regularization loss $\mathcal{L}_{\text{reg}}$ regularizes depth at seen viewpoints.

To achieve this, we propose a novel regularization technique to enforce a geometric consistency across different views with depth-guided warping and consistency modeling. We focus on the fact that NeRF Mildenhall et al. (2020) inherently renders not only color image but depth image as well. Combined with known viewpoint difference, the rendered depths can be used to define a geometric correspondence relationship between two arbitrary views.

Specifically, we consider a depth image rendered by the NeRF model, $D_j$ at unseen viewpoint $j$. By formulating a warping function $\psi(I_i; D_j, R_{i \to j})$ that warps an image $I_i$ according to the depth $D_j$ and viewpoint difference $R_{i \to j}$, we can encourage a consistency between warped image $I_{i \to j} = \psi(I_i; D_j, R_{i \to j})$ and rendered image $I_j$ at $j$-th unseen viewpoint, which in turn improves the few-shot novel view synthesis performance. This framework can overcome the limitations of previous few-shot setting approaches Mildenhall et al. (2020); Chen et al. (2021); Barron et al. (2021), improving not only global geometry but also high-frequency details and appearance as well.

In the following, we first explain how input images can be warped to unseen viewpoints in our framework. Then, we demonstrate how we impose consistency upon the pair of warped image and rendered image for regularization, followed by explanation of occlusion handling method and several training strategies that proved crucial for stabilization of NeRF optimization in few-shot scenario.

## 4.2 RENDERED DEPTH-GUIDED WARPING

To render an image at novel viewpoints, we first sample a random camera viewpoint, from which corresponding ray vectors are generated in a patch-wise manner. As NeRF outputs density and color values of sampled points along the novel rays, we use recovered density values to render a consistent depth map. Following Mildenhall et al. (2020), we formulate per-ray depth values as weighted composition of distances traveled from origin. Since ray $\mathbf{r}_p$ corresponding to pixel $p$ is parameterized as $\mathbf{r}_p(t) = \mathbf{o} + t\mathbf{d}_p$, the depth rendering is defined similarly to the color rendering:

$$D(\mathbf{r}_p) = \int_{t_n}^{t_f} T(t)\sigma(\mathbf{r}_p(t))t\,dt, \tag{4}$$

where $D(\mathbf{r}_p)$ is a predicted depth along the ray $\mathbf{r}_p$. As described in Figure 1, we use the rendered depth map $D_j$ to warp input ground truth image $I_i$ to $j$-th unseen viewpoint and acquire a warped image $I_{i \to j}$, which is defined as a process such that $I_{i \to j} = \psi(I_i; D_j, R_{i \to j})$. More specifically, pixel location $p_j$ in target unseen viewpoint image is transformed to $p_{j \to i}$ at source viewpoint image by viewpoint difference $R_{j \to i}$ and camera intrinsic parameter $K$ such that

$$p_{j \to i} \sim K R_{j \to i} D_j(p_j) K^{-1} p_j, \tag{5}$$

where $\sim$ indicates approximate equality and the projected coordinate $p_{j \to i}$ is a continuous value. With a differentiable sampler, we extract color values of $p_{j \to i}$ on $I_i$. More formally, the transforming components process can be written as follows:

$$I_{i \to j}(p_j) = \text{sampler}(I_i; p_{j \to i}), \tag{6}$$

where $\text{sampler}(\cdot)$ is a bilinear sampling operator Jaderberg et al. (2015).

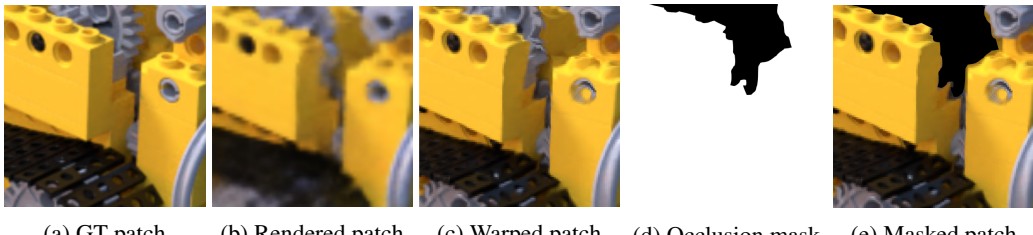

| (a) GT patch | (b) Rendered patch | (c) Warped patch | (d) Occlusion mask | (e) Masked patch |

Figure 3: **Visualization of consistency modeling process.** (a) ground truth patch, (b) rendered patch at novel viewpoint, (c) warped patch, from input viewpoint to novel viewpoint, (d) occlusion mask with threshold masking, and (e) final warped patch with occlusion masking at novel viewpoint.

**Acceleration.** Rendering full image with NeRF voluemtric rendering is computationally heavy and extremely timetaking, requiring tens of seconds for a single iteration. To overcome the computational bottleneck of full image rendering and warping, rays are sampled on a strided grid to make the patch with stride $s$, which we have set as 2. After the rays undergo volumetric rendering, we upsample the low-resolution depth map back to original resolution with bilinear interpolation. This full-resolution depth map is used for the inverse warping. This way, detailed warped patches of full-resolution can be generated with only a fraction of computational cost that would be required when rendering the original sized ray batch.

### 4.3 CONSISTENCY MODELING

Given the rendered patch $I_j$ at $j$-th viewpoint and the warped patch $I_{i \to j}$ with depth $D_j$ and viewpoint difference $R_{i \to j}$, we define the consistency between the two to encourage additional regularization for globally consistent rendering. One viable option is to naïvely apply the pixel-wise image reconstruction loss $\mathcal{L}_{\text{pix}}$ such that

$$\mathcal{L}_{\text{pix}} = \|I_{i \to j} - I_j\|. \tag{7}$$

However, we observe that this simple strategy is prone to cause failures in reflectant non-Lambertian surfaces where appearance changes greatly regarding viewpoints Zhan et al. (2018). In addition, geometry-related problems, such as self-occlusion and artifacts, prohibits naïve usage of pixel-wise image reconstruction loss for regularization in unseen viewpoints.

**Feature-level consistency modeling.** To overcome these issues, we propose masked feature-level regularization loss that encourages structural consistency while ignoring view-dependent radiance effects, as illustrated in Figure 2.

Given an image $I$ as an input, we use a convolutional network to extract multi-level feature maps such that $f_{\phi,l}(I) \in \mathbb{R}^{H_l \times W_l \times C_l}$, with channel depth $C_l$ for $l$-th layer. To measure feature-level consistency between warped image $I_{i \to j}$ and rendered image $I_j$, we extract their features maps from $L$ layers and compute difference within each feature map pairs that are extracted from the same layer.

In accordance with the idea of using the warped image $I_{i \to j}$ as pseudo ground truths, we allow a gradient backpropagation to pass only through the rendered image and block it for the warped image. By applying the consistency loss at multiple levels of feature maps, we cause $I_j$ to model after $I_{i \to j}$ both on semantic and structural level.

Formally written, the consistency loss $\mathcal{L}_{\text{cons}}$ is defined as such that,

$$\mathcal{L}_{\text{cons}} = \sum_{l=1}^{L} \frac{1}{C_l} \|f_{\phi}^l(I_{j \to i}) - f_{\phi}^l(I_j)\|. \tag{8}$$

For this loss function $\mathcal{L}_{\text{cons}}$, we find $l$-1 distance function most suited for our task and utilize it to measure consistency across feature difference maps. Empirically, we have discovered that VGG-19 network Simonyan & Zisserman (2014) yields best performance in modeling consistencies, likely due to the absence of normalization layers Johnson et al. (2016) that scale down absolute values of feature differences. Therefore, we employ VGG19 network as our feature extractor network $f_{\phi}$ throughout all of our models.

It should be noted that our loss function differs from that of DietNeRF Jain et al. (2021) in that while DietNeRF's consistency loss is limited to regularizing the radiance field in a globally semantic level,

our loss combined with warping module is also able to give the network highly rich information on a local, structural level as well. In other words, contrary to DietNeRF giving only high-level feature consistency, our method of using multiple levels of convolutional network for feature difference calculation can be interpreted as enforcing a mixture of all levels, from high-level semantic consistency to low-level structural consistency.

**Occlusion handling.** In order to prevent imperfect and distorted warpings caused by erroneous geometry from influencing the model, which degrade overall reconstruction quality, we construct consistency mask $M_l$ to let NeRF ignore regions with geometric inconsistencies, as demonstrated in Figure 3. Instead of applying mask to the images before inputting them into feature extractor network, we apply resized masks $M_l$ directly to the feature maps, after using nearest-neighbor down-sampling to make them match the dimensions of $l$-th layer outputs.

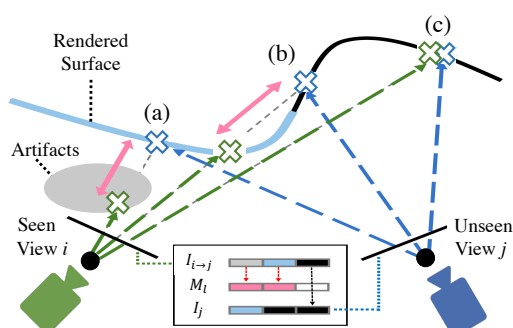

Figure 4: **Occlusion-aware mask generation.** Mask generation by comparing geometry between novel view $j$ and source view $i$, with $I_{i \rightarrow j}$ being warped patch generated for view $j$. For (a) and (b), warping does not occur correctly due to artifacts and self-occlusion, respectively. Such pixels are masked out by $M_l$, allowing only (c), with accurate warping, as training signal for rendered image $I_j$.

We generate $M$ by measuring consistency between rendered depth values from target viewpoint and source viewpoint such that

$$M(p_j) = \big[\|D_j(p_j) - D_i(p_{j \rightarrow i})\| < \tau \big]. \quad (9)$$

where $[\cdot]$ is Iverson bracket, and $p_{j \rightarrow i}$ refers to the corresponding pixel in source viewpoint $i$ for reprojected target pixel $p_j$ of $j$-th viewpoint. Here we measure euclidean distance between depth points rendered from target and source viewpoints as a criterion for a threshold masking. As illustrated in Figure 4, if distance between two points are greater than given threshold value $\tau$, we determine two rays as rendering depths of separate surfaces and mask out the corresponding pixel in viewpoint $I_j$. The process takes place over every pixel in viewpoint $I_j$ to generate a mask $M$ the same size as rendered pixels. Through this technique, we filter out problematic solutions at feature level and regularize NeRF with only high-confidence image features.

Based on this, the consistency loss $\mathcal{L}_{\text{cons}}$ is extended as such that

$$\mathcal{L}_{\text{cons}}^M = \sum_{l=1}^{L} \frac{1}{C_l m_l} \big\| M_l \odot (f_\phi^l(I_{i \rightarrow j}) - f_\phi^l(I_j)) \big\|, \quad (10)$$

where $m_l$ is the sum of non-zero values.

**Edge-aware disparity regularization.** Since our method is dependent upon the quality of depth rendered by NeRF, we directly impose additional regularization on rendered depth to facilitate optimization. We further encourage local depth smoothness on rendered scenes by imposing $l$-1 penalty on disparity gradient within randomly sampled patches of input views. In addition, inspired by Godard et al. (2017), we take into account the fact that depth discontinuities in depth maps are likely to be aligned to gradients of its color image, and introduce an edge-aware term with image gradients $\partial I$ to weight the disparity values. Specifically, following Godard et al. (2017), we regularize for edge-aware depth smoothness such that

$$\mathcal{L}_{\text{reg}} = |\partial_x D_i^*|e^{-|\partial_x I_i|} + |\partial_y D_i^*|e^{-|\partial_y I_i|}, \quad (11)$$

where $D_i^* = D_i / \overline{D_i}$ is the mean-normalized inverse depth from Godard et al. (2017) to discourage shrinking of the estimated depth.

## 4.4 TRAINING STRATEGY

In this section, we present novel training strategies to learn the model with the proposed losses.

**Total losses.** We optimize our model with a combined final loss of original NeRF's pixel-wise reconstruction loss $\mathcal{L}_{\text{obs}}$ and two types of regularization loss, $\mathcal{L}_{\text{cons}}^M$ for unobserved view consistency modeling and $\mathcal{L}_{\text{reg}}$ for disparity regularization.

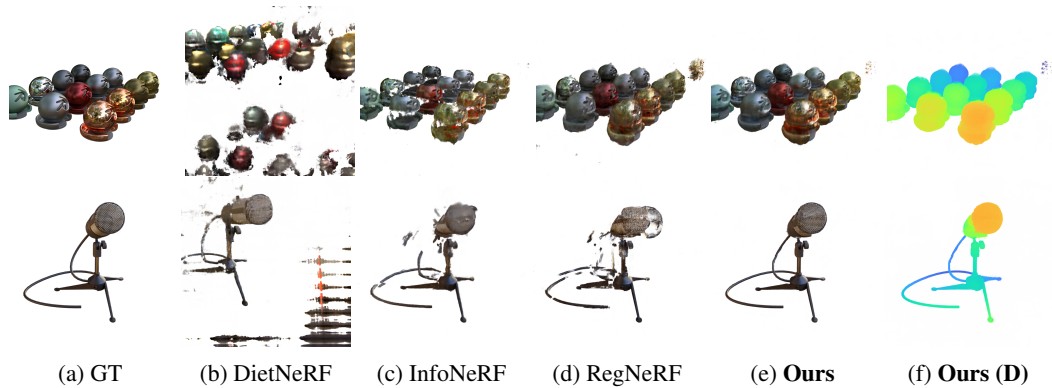

|  (a) GT | (b) DietNeRF | (c) InfoNeRF | (d) RegNeRF | (e) **Ours** | (f) **Ours (D)** |

Figure 5: **Qualitative comparison on NeRF-Synthetic Mildenhall et al. (2020)** show that in 3-view setting, our method captures fine details more robustly (such as the wire in the *mic* scene) and produces less artifacts (background in the *materials* scene) compared to previous methods. We show GeCoNeRF's results (e) with its rendered depth (f).

**Progressive camera pose generation.** Difficulty of of accurate warping increases the further target view is from the source view, which means that sampling far camera poses straight from the beginning of training may have negative effects on our model. Therefore, we first generate camera poses near source views, then progressively further as training proceeds. We sample noise value uniformly between an interval of $[-\beta, +\beta]$ and add it to the original Euler rotation angles of input view poses, with parameter $\beta$ growing linearly from 3 to 9 degrees throughout the course of optimization. This design choice can be intuitively understood as stabilizing locations near observed viewpoints at start and propagating this regularization to further locations, where warping becomes progressingly more difficult.

**Positional encoding frequency annealing.** We find that most of the artifacts occurring are high-frequency occlusions that fill the space between scene and camera. This behaviour can be effectively suppressed by constraining the order of fourier positional encoding Tancik et al. (2020) to low dimensions. Due to this reason, we adopt coarse-to-fine frequency annealing strategy previously used by Park et al. (2021) to regularize our optimization. This strategy forces our network to primarily optimize from coarse, low-frequency details where self-occlusions and fine features are minimized, easing the difficulty of warping process in the beginning stages of training. Following Park et al. (2021), the annealing equation is $\alpha(t) = mt/K$, with $m$ as the number of encoding frequencies, $t$ as iteration step, and we set hyper-parameter $K$ as $15k$.

## 5 EXPERIMENTS

### 5.1 EXPERIMENTAL SETTINGS

**Baselines.** We use mip-NeRF Barron et al. (2021) as our backbone. We give our comparisons to the baseline and several state-of-the-art models for few-shot NeRF: InfoNeRF Kim et al. (2022), DietNeRF Jain et al. (2021), and RegNeRF Niemeyer et al. (2022).

**Datasets and metrics.** We evaluate our model on NeRF-Synthetic Mildenhall et al. (2020) and LLFF Mildenhall et al. (2019). NeRF-Synthetic is a realistically rendered 360° synthetic dataset comprised of 8 scenes. We randomly sample 3 viewpoints out of 100 training images in each scene, with 200 testing images for evaluation. We also conduct experiments on LLFF benchmark dataset, which consists of real-life forward facing scenes. Following RegNeRF Niemeyer et al. (2022), we apply standard settings by selecting test set evenly from list of every 8th image and selecting 3 reference views from remaining images. We quantify novel view synthesis quality using PSNR, Structural Similarity Index Measure (SSIM) Wang et al. (2004), LPIPS perceptual metric Zhang et al. (2018) and "average" error metric introduced in Barron et al. (2021) to report the mean value of metrics for all scenes in each dataset.

**Implementation details.** Our main model is built on top of the JAX mip-NeRF codebase Barron et al. (2021). We use Adam optimizer using an exponential learning rate decay. Our model is trained for 60k iterations for 4 hours on two NVIDIA RTX3090Ti GPUs. We provide more implementation details in supplementary materials.

Table 1: **Quantitative comparison on NeRF-Synthetic (Mildenhall et al., 2020) and LLFF (Mildenhall et al., 2019) datasets.**

| Methods | NeRF-Synthetic (Mildenhall et al., 2020) | | | | LLFF (Mildenhall et al., 2019) | | | |
|---|---|---|---|---|---|---|---|---|
| | PSNR ↑ | SSIM ↑ | LPIPS ↓ | Avg. ↓ | PSNR ↑ | SSIM ↑ | LPIPS ↓ | Avg. ↓ |
| NeRF (Mildenhall et al., 2020) | 14.73 | 0.734 | 0.451 | 0.199 | 13.34 | 0.373 | 0.451 | 0.255 |
| mip-NeRF (Barron et al., 2021) | 17.71 | 0.798 | 0.745 | 0.178 | 14.62 | 0.351 | 0.495 | 0.246 |
| DietNeRF (Jain et al., 2021) | 16.06 | 0.793 | 0.306 | 0.151 | 14.94 | 0.370 | 0.496 | 0.232 |
| InfoNeRF (Kim et al., 2022) | 18.65 | 0.811 | 0.230 | 0.111 | 14.37 | 0.349 | 0.457 | 0.238 |
| RegNeRF (Niemeyer et al., 2022) | 18.01 | 0.842 | 0.747 | 0.167 | 19.08 | 0.587 | 0.336 | 0.146 |
| **GeCoNeRF (Ours)** | **19.23** | **0.866** | **0.723** | **0.148** | **18.55** | **0.578** | **0.340** | **0.150** |

| (a) Ground-truth | (b) mip-NeRF | (c) mip-NeRF (D) | (d) **Ours** | (e) **Ours (D)** |

Figure 6: **Qualitative results on LLFF Mildenhall et al. (2019).** Comparison with baseline mip-NeRF shows that our model learns of coherent depth and geometry in extremely sparse 3-view setting.

## 5.2 COMPARISONS

**Qualitative comparisons.** Qualitative comparison results in Figure 5 and 6 demonstrate that our model shows superior performance to baseline mip-NeRF Barron et al. (2021) and previous state-of-the-art model, RegNeRF Niemeyer et al. (2022), in 3-view settings. We observe that our warping-based consistency enables GeCoNeRF to capture fine details that mip-NeRF and RegNeRF struggle to capture in same sparse view scenarios, as demonstrated with the *mic* scene. Our method also displays higher stability in rendering smooth surfaces and reducing artifacts in background in comparison to previous models, as shown in the results of the *materials* scene. We argue that these results demonstrate how our method, through generation of warped pseudo ground truth patches, is able to give the model local, scene-specific regularization that aids recovery of fine details, which previous few-shot NeRF models with their global, generalized priors were unable to accomplish.

**Quantitative comparisons.** Comparisons in Table 1 shows our model's competitive results in LLFF dataset, whose PSNR results show large increase in comparison to mip-NeRF baseline and competitive compared to RegNeRF. We see that our warping-based consistency modeling successfully prevents overfitting and artifacts, which allows our model to perform better quantitatively.

## 5.3 ABLATION STUDY

We validate our design choices by performing an ablation study on LLFF Mildenhall et al. (2019) dataset. Quantitative and qualitative results are given in Table 2 and Figure 7, respectively.

Table 2: **Ablation study.**

| Components | PSNR↑ | SSIM↑ | LPIPS↓ | Avg.↓ |
|---|---|---|---|---|
| **(a)** Baseline | 14.62 | 0.351 | 0.495 | 0.246 |
| **(b)** (a) + $\mathcal{L}_{\text{cons}}$ | 18.10 | 0.529 | 0.408 | 0.164 |
| **(c)** (b) + $M$ (O. mask) | 18.24 | 0.535 | 0.379 | 0.159 |
| **(d)** (c) + Progressive | 18.46 | 0.552 | 0.349 | 0.151 |
| **(e)** (d) + $\mathcal{L}_{\text{reg}}$ **(Ours)** | **18.55** | **0.578** | **0.340** | **0.150** |

**Feature-level consistency loss.** We observe that without the consistency loss $\mathcal{L}_{\text{cons}}$, our model suffers both quantitative and qualitative decrease in reconstruction fidelity, verified by incoherent geometry in image (a) of Figure 7. Absence of unseen view consistency modeling destabilizes the model, resulting divergent behaviours such as artifact generation in empty space.

**Occlusion mask.** We observe that addition of occlusion mask $M$ improves overall appearance as well as geometry, as shown in image (c) of Figure 7. Its absence results broken geometry throughout the overall scene, as demonstrated in (b). Erroneous artifacts pertaining to projections from different viewpoints were detected in multiple scenes, resulting lower quantitative values.

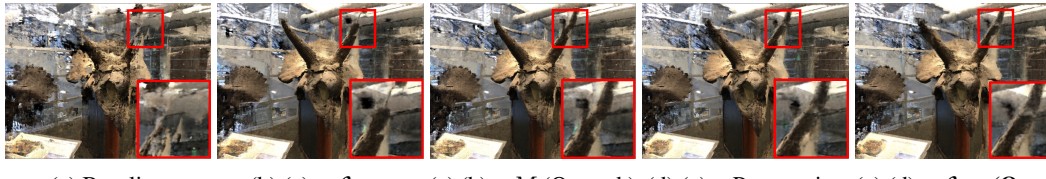

(a) Baseline        (b) (a) + $\mathcal{L}_{\text{cons}}$    (c) (b) + $M$ (O. mask)  (d) (c) + Progressive  (e) (d) + $\mathcal{L}_{\text{reg}}$ **(Ours)**

Figure 7: **Qualitative ablation.** Our qualitative ablation results on *Horns* scene shows the contribution of each module in performance of our model at 3-view scenario.

**Progressive training strategies.** In Table 3, we justify our progressive training strategies with additional experiments on NeRF-Synthetic dataset, while in the main ablation we conduct an ablation with progressive annealing only. For pose generation, we sample pose angle from large interval in the beginning, instead of slowly growing the interval. For positional encoding, we replace progressive annealing with naïve positional encoding used in NeRF. We observe that their absence causes destabilization of the model and degradation in appearance, respectively.

**Feature-level loss vs. pixel-level loss.** In Table 4, we conduct a quantitative ablation comparisons between feature-level consistency loss $\mathcal{L}_{\text{cons}}$ and pixel-level photometric consistency loss $\mathcal{L}_{\text{pix}}$, both with occlusion masking. As shown in Figure 8, naïvely applying pixel-level loss for consistency modeling leads to broken geometry. This phenomenon can be attributed to $\mathcal{L}_{\text{pix}}$ being agnostic to view-dependent specular effects, which the network tries to model by altering or erasing altogether non-Lambertian surfaces.

Table 3: **Progressive training ablation.**

| Components | PSNR↑ | SSIM↑ | LPIPS↓ | Avg. ↓ |
|---|---|---|---|---|
| w/o prog. anneal | 18.50 | 0.852 | 0.781 | 0.161 |
| w/o prog. pose | 16.96 | 0.799 | 0.811 | 0.194 |
| w/o both | 17.04 | 0.788 | 0.823 | 0.197 |
| GeCoNeRF (Ours) | **19.23** | **0.866** | **0.723** | **0.148** |

Table 4: **Pixel-level consistency ablation.**

| Components | PSNR↑ | SSIM↑ | LPIPS↓ | Avg.↓ |
|---|---|---|---|---|
| w/ $\mathcal{L}_{\text{pix}}$ | 17.98 | 0.528 | 0.431 | 0.165 |
| w/ $\mathcal{L}_{\text{cons}}$ | **18.55** | **0.578** | **0.340** | **0.150** |

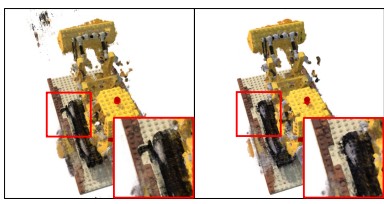

(a) Pixel-level        (b) Feature-level

Figure 8: $\mathcal{L}_{\text{pix}}$ **vs.** $\mathcal{L}_{\text{cons}}$ **comparison.**

### 5.4 COMPARISON TO CONSISTENCY MODELING BETWEEN KNOWN VIEWS

In order to compare GeCoNeRF with contemporary methods Darmon et al. (2022); Chen et al. (2022); Fu et al. (2022) that model consistency between known views, we conduct an experiment to observe how such consistency modeling performs in few-shot NeRF setting. In our experiment, we replace our consistency modeling with above setting, warping source images to other known views for consistency between the warped image and ground truth (reference) image.

Its result, shown in (a) of Figure 9, displays divergent behaviours such as heavy artifact generation, while our method (b) succeeds in recovering detailed geometry of the scene under the same setting. As discussed in Section 2, we argue that large view differences and scarcity of reference images make it difficult for NeRF to refine geometry with consistency modeling between known views. Our work's novel contributions allow consistency modeling to be adopted to few-shot NeRF to facilitate stable training under such extreme conditions, distinguishing our work from above methods.

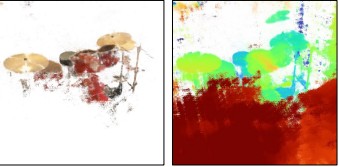

(a) Between known views

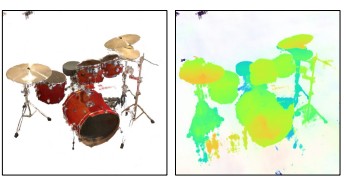

(b) GeCoNeRF (Ours)

Figure 9: **Consistency between known views vs. our method.**

## 6 CONCLUSION

We present GeCo-NeRF, a novel few-shot NeRF regularization method. We regularize geometry by modeling feature-level consistency at unobserved viewpoints between using the warped images, regularizing NeRF for learning of robust geometry. Further techniques and training strategies we propose prove to have stabilizing effect and facilitate optimization of our network. Our experimental evaluation demonstrates our method's competitiveness in regards to other state-of-the-art models.

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
