# OpenReview forum: "Neural Radiance Fields with Geometric Consistency for Few-Shot Novel View Synthesis"
_ICLR.cc/2023/Conference — Submitted to ICLR 2023_

### Official Review · Reviewer_Copi · 2022-10-13

**Confidence:** 4
**Correctness:** 4
**Technical Novelty And Significance:** 2
**Empirical Novelty And Significance:** 3
**Recommendation:** 8

**Clarity, Quality, Novelty And Reproducibility:**

The paper is well-written and clearly demonstrated its contribution. The results presented both qualitatively and quantitatively outperforms baselines.

The paper should be reproduced with some efforts according to the details provided. Due to the several training strategies proposed, it might be hard to achieve the same results as in the paper. Therefore, it would be great if the code can be open-sourced after publication.

**Strength And Weaknesses:**

Strength:
- The results are quite promising and achieve the state-of-the-art in the few-shot NeRF setting
- The authors put a lot of engineering efforts to make the warping technique work

Weaknesses:
- The method is not novel enough for me. Many methods propose to supervise at unseen viewpoints, e.g. DietNeRF.  Also, in [1], input image color was used to supervise unseen viewpoints at pixel level.
- The authors doesn't discuss the computational cost of the proposed method. In the paper, they report that 8 hours is needed to train 120k iterations with 2 RTX 3090Ti. As far as I know, training vanilla NeRF for 200k iterations only need 8 hours on a single 1080Ti.
- Unless I am missing something, ablation is not provided for the training strategy in sec 4.5, which I think is part of the core contribution of this paper
- Some references can be incorporated and discussed: concurrent work [2], [3] also uses estimated depth to warp input images to other camera poses to regularize NeRF (This point doesn't influence my rating, but including them would make this paper more complete)
- It seems that I cannot find the mentioned supplementary material

Minor:
- some references are not compiled successfully, as can be seen in sec 5.1
- In "Plugging in NeRF'' in sec 5.3, table 3 should be table 1

[1] Ray Priors through Reprojection: Improving Neural Radiance Fields for Novel View Extrapolation

[2] Data augmentation for NeRF: a geometric consistent solution based on view morphing

[3] StructNeRF: Neural Radiance Fields for Indoor Scenes with Structural Hints

**Summary Of The Paper:**

This paper focus on the novel view synthesis with only sparse inputs. The authors propose a method called GeCo-NeRF that enforces geometric consistency to regularize the training of NeRF. Specifically, input images are warped to unseen viewpoints for supervision at feature level. They also filter out erroneous warped patches and engineer several training strategies to improve training. Experiments on both synthetic and real data demonstrated the effectiveness of GeCo-NeRF.

**Summary Of The Review:**

Overall, I think this paper falls into the borderline. The techniques are not novel enough and some details are missing, but I appreciate the engineering effort and the results are quite promising. Therefore, I lean towards acceptance.

---

> ### Author Response · Authors · 2022-11-16
> **Official Comment to Reviewer Copi (2/2)**
>
> ### C3. Lack of training strategy ablation
>
> **A3.** Thank you for pointing this out! As you have said, we have added a section for ablation experiments of progressive training strategy in the revised version of our paper, as it is part of our core contributions. We demonstrate how the absence of progressive pose generation and progressive annealing destabilizes the model and reduces the reconstruction quality, respectively. We apologize that it had not been included in the previous ablation experiments.
>
> &nbsp;
>
> ### C4. Additional references
>
> **A4.** We are thankful for your valuable suggestion. We agree that including the works you have mentioned would make our paper more complete and thus have included all the mentioned works in the revised version of our paper. Please check the newly made section in our Related Work named “**Self-supervised photometric consistency”**, where we introduce and compare the works you have mentioned in detail.
>
> &nbsp;
>
> ### C5. Regarding supplementary material
>
> **A5.** We apologize for not including the supplementary material in our first submission, as it was a mistake on our part. In the revised paper, we have included the mentioned supplementary materials to the full. Thank you for your thorough revision of our work.
>
> &nbsp;
>
> ### C6. Open-sourced Code
>
> **A6.** We agree wholeheartedly with your comment. We promise to open-source the code after publication, as it may not be easy to recreate our results due to the number of engineering details involved. Thank you for your suggestion.
>
> &nbsp;
>
> We hope our responses have addressed the concerns of the reviewer and we are happy to answer any further questions.
>
> &nbsp;
>
> ### **References**
>
> [1] Jain, Ajay, et al. “Putting nerf on a diet: Semantically consistent few-shot view synthesis”, ICCV 2021
>
> [2] Kim, Mijeong, et al. “InfoNeRF: Ray Entropy Minimization for Few-Shot Neural Volume Rendering”, CVPR 2022
>
> [3] Niemeyer, Michael, et al. “RegNeRF: Regularizing Neural Radiance Fields for View Synthesis from Sparse Inputs,” CVPR 2022
>
> [4] Zhang, Jian, et al. “Ray Priors through Reprojection: Improving Neural Radiance Fields for Novel View Extrapolation,” CVPR 2022

---

> ### Author Response · Authors · 2022-11-16
> **Official Comment to Reviewer Copi (1/2)**
>
> **General reply:** Thank you for your constructive review and helpful suggestions! We give a detailed response to your questions and comments in the following. If any of our responses do not adequately address your concerns, please let us know and we will get back to you as soon as possible.
>
> &nbsp;
>
> ### C1. Regarding novelty
>
> **A1.** Thank you for your comment. We would like to clarify that regularization of NeRF at unseen viewpoints is not our novelty, as we were fully aware how all of the methods that we mentioned in “Few-shot NeRF” section of Related Work, such as DietNeRF [1], InfoNeRF [2] and RegNeRF [3] all regularize NeRF at unseen viewpoints. However, their regularization only operate on global and generalized scale to make the scene geometry and appearance more “realistic” in general.
>
> We believe our main contribution, which differentiates our work from previous works, is that we demonstrate how NeRF can be regularized in **local**, **fine**, and **scene-specific** manner with usage of warped images in unseen viewpoints and multi-level feature consistency. This enables our model to recover fine details that previous methods are unable to, as demonstrated by experimental results in Figure 5 and explained more thoroughly in **“Qualitative comparisons”** in Section 5.2 of our revised work.
>
> As for RapNeRF [4], we would like to point out the some differences between our paper:
>
> - RapNeRF does not address few-shot neural radiance fields but starts with the assumption that dense, numerous viewpoints are given. Its training details reveal that its procedure is actually a method of fine-tuning from pretrained NeRF to refine the reconstruction quality at unseen viewpoints. Our method focuses on few-shot setting and demonstrates that using warped images in unobserved viewpoints for consistency modeling serves as a strong regularization that stabilizes NeRF’s training in sparse view setting, something which no other paper has done before.
> - Unlike RapNeRF which naively uses reprojected pixel values as ground truth, we introduce feature-level consistency to let the model learn multi-level perceptual similarity, preventing the model from overfitting to source images’ colors (and thus suppressing specular effects) in unseen viewpoints - we see that it resolves the main weakness that RapNeRF mentions in its abstract.
> - RapNeRF does not address self-occlusions, which are bound to occur when warping between viewpoints. In our work, we introduce a novel occlusion handling method to filter out erroneous geometry and use only accurately warped pixels as pseudo ground truth.
>
> &nbsp;
>
> ### C2. Computational cost
>
> **A2.**  As you have stated, it is true that the computational cost of our method is greater than that of NeRF, making it slower. There are two major reasons for this: first reason being warping module taking up computational cost every iteration, and the second reason being our occlusion handling procedure. Because we have to render a point from both unseen view and seen view every iteration to generate the mask, it takes up twice as much computational cost to render a single ray in our method. We prevent gradient flow from flowing through the mask, so the computation regarding model optimization remains the same - however, each iteration step becomes twice as slower. We will make sure to address the computational cost of our method in detail in final version of our paper. Also, we can address in future work how this computational cost can be reduced - for example, one solution can be replacing our occlusion handling method with an algorithm that compares reprojection order using depth. Thank you for your constructive suggestion.

---

### Official Review · Reviewer_8ZEs · 2022-10-23

**Confidence:** 4
**Correctness:** 3
**Technical Novelty And Significance:** 2
**Empirical Novelty And Significance:** 2
**Recommendation:** 5

**Clarity, Quality, Novelty And Reproducibility:**

In terms of writing, the paper is organized, and the overall flow is clear. However, there are a few typos in the paper. For example, in Figure 1 illustration, it should be NeRF instead of NerF. Some citations with question marks (?) in the paper.

Some idea needs more clarification, for example, in section 2, "... the density’s entropy in each ray and ensures consistency across rays in the neighborhood. However, the performance of these methods was still limited". It should be discussed in-depth that why their performance is limited, as these methods are direct competitors to the proposed approach. In section 4.4, quote “As explained above, we do not … generating Ml”. I think the authors try to explain that they don’t apply masks directly to the image pairs, but to the feature map level. However, the sentence needs to be reorganized.


**Strength And Weaknesses:**

Strengths:

1. A novel way to generate pseudo ground truth by warpping multiple input views to the novel view location. The local features are aligned using NeRF generated depth map by an occlusion-aware mask map.

2. Instead of vanilla NeRF only using pixel-level reconstruction loss, the paper proposes to add multi-level perceptual loss between wrapped pseudo ground truth image and generated novel view image. By adding such constrain at the feature level, the framework improves quality of generated novel views by a large margin in comparison to the baselines.

Weaknesses:

1. The novelty is not significant. The main contribution of this paper is the depth rewarping loss. However, using depth to for few-shot nerf is not new [1][2]. Also, warping input to unseen views is widely used in self-supervised MVS [3][4].

2. It is not convincing that depth estimation (Eq. 4) is accurate enough with few viewpoints. Also, the resolution could be limited given the fact that the depth is generated with grid sampling.

3. It is weird to mention Monocular depth estimation in related work, which may be self-supervised photometric consistency.

4. There are a lot of missing citations.

[1] Depth-supervised NeRF: Fewer views and faster training for free

[2] Dense depth priors for neural radiance fields from sparse input views.

[a] M3vsnet: Unsupervised multi-metric multi-view stereo network

[b] Learning Unsupervised Multi-View Stereopsis via Robust Photometric Consistency

**Summary Of The Paper:**

This paper proposes a method that regularize nerf in a few-shot setting. The method utilize NeRF rendered depth as pseudo ground truth to warp rgb value to other near view. The error between source view and target view are taken as an additional loss to train NeRF in a few shot setting.

**Summary Of The Review:**

Based on the weakness discussed above, I think the paper has limited novelty, and there is also some defacts in terms of technical and writing issues. Therefore, I vote for rejection at this stage.

---

> ### Author Response · Authors · 2022-11-16
> **Official Comment to Reviewer 8ZEs (2/2)**
>
> ### C2. Accuracy of depth estimation with few viewpoints
>
> **A2.** We understand your concern and acknowledge that the depth recovered by a few-shot NeRF is far from perfect, especially when it comes to fine details. However, we would like to address your comment in two parts:
>
> **Estimated depth accuracy**
>
> We would like to clarify that we find coarse geometry recovered by NeRF in the few-shot setting, while not perfect, is sufficient for providing enough **correspondence information** for warping to occur coherently between source images and nearby random viewpoints. Our ablation results support this claim. We also point out that we add multiple constraints such as disparity regularization (Section 4.3), occlusion handling (Section 4.3) and progressive annealing (Section 4.4) to aid NeRF in recovering more coherent geometry despite sparse view settings, all of which we demonstrate as effective in our experiments.
>
> **Limited resolution**
>
> As for the resolution, it is not limited, as it is trivial to sample more rays from NeRF to generate depth images of unlimited resolutions. As described in Section 4.2 of our paper, it is only due to the computational cost required for backpropagation and the speed of our training that we have restricted the resolution of our depth images with grid sampling during the training procedure. For inference, however, we point out that we are not bound by this limitation in any manner.  However, we acknowledge that we may have not been clear enough in our explanations, and inspired by your comment we have added a new subsection that clarifies grid sampling-related details, named **“Acceleration”**, in Section 4.2 of our revised paper.
>
> &nbsp;
>
> ### C3. Regarding subsection “Monodepth estimation” in Related Work
>
> **A3.** Thank you for your suggestion. Following it, we have removed the subsection **“Monodepth estimation”** from our Related Work and added a new subsection named **“Self-supervised Photometric Consistency”**, where we introduce and explain the works that use consistency modeling. We have included all the references you have mentioned in our revised Related Work, and we would be grateful if you could check.
>
> &nbsp;
>
> ### C4. Missing citations
>
> **A4.** We are sincerely grateful for your careful reading of our work and for pointing out these mistakes. We have fixed all typos and erroneous references in the revised version of our paper.
>
> &nbsp;
>
> ### Response to *Clarity, Quality, Novelty and Reproducibility*:
>
> **A5.** **Typos and wrong citations.** As we have stated in **A4**, we have fixed all typos and wrong citations in the revised version of our paper. We would be grateful if you could check it.
>
> **A5. Clarification of sentence in Section 2**. We completely agree. We have modified the given section in the revised version of our paper to emphasize how the competitive methods are limited in comparison to ours. As summarized in the paper, their main limitation comes from their usage of global & handcrafted priors for regularization: such priors require extensive fine-tuning of parameters for each dataset and only provide regularization at a global scale in a generalized manner. However, as demonstrated in Figure 5 of our work, our warping-based consistency regularization propagates information at a more local, scene-specific scale which allows our method to recover fine geometry details that previous methods were not able to reconstruct.
>
> **A6. Sentence reorganization.** Thank you for your suggestion: we agree that some of our sentences were not well organized and were misleading in some aspects. We have fixed and re-written all such sentences. Also, following your comment, we judged that our explanation of occlusion mask generation was unclear overall, so we have fixed much of the section as well as added **Figure 4** regarding mask generation to make our methods easier to understand. We would appreciate it greatly if you could review it.
>
> &nbsp;
>
> We hope our responses have addressed the concerns of the reviewer and we are happy to answer any further questions.
>
> &nbsp;
>
> ### **References**
>
> [1] Deng, Kangle, et al. “Depth-supervised NeRF: Fewer views and faster training for free”, CVPR 2022
>
> [2] Roessle, Barbara, et al. “Dense depth priors for neural radiance fields from sparse input views”, CVPR 2022
>
> [3] Huang, Baichuan, et al. “M3vsnet: Unsupervised multi-metric multi-view stereo network”, ICIP 2021
>
> [4] Khot, Tajas, et al. “Learning Unsupervised Multi-View Stereopsis via Robust Photometric Consistency”, ArXiv 2019

---

> > ### Comment · Reviewer_8ZEs · 2022-11-24
> > **Feedback**
> >
> > Thanks the authors for the elaborate response. Although I am still not totally convinced about the novelty issue, they've addressed most of my concerns, and the revised version looks better. Therefore, I'll raise my score.

---

> > > ### Author Response · Authors · 2022-11-24
> > > **Thank you**
> > >
> > > Dear Reviewer 8ZEs,
> > >
> > > We would like to thank you again for your constructive review. Your review really helped us greatly in improving our paper and we are truly grateful for your comments. We are very happy to see that our revision has addressed your concerns. We sincerely appreciate your suggestions.
> > >
> > > Thank you!
> > >
> > > Paper765 Authors

---

> > > ### Author Response · Authors · 2022-11-26
> > > **A friendly reminder**
> > >
> > > Dear Reviewer 8ZEs,
> > >
> > > We thank you once more for your positive comment. We would like to friendly remind you that the recommendation score in the official review is not changed yet, and we sincerely hope that you could officially edit it at your convenience.
> > >
> > > We are grateful for your time and effort in reviewing our paper and reading our comments.
> > >
> > > Paper 765 Authors

---

> ### Author Response · Authors · 2022-11-16
> **Official Comment to Reviewer 8ZEs (1/2)**
>
> **General reply:** Thank you for your constructive review and helpful suggestions! We give a detailed response to your questions and comments in the following. If any of our responses do not adequately address your concerns, please let us know and we will get back to you as soon as possible.
>
> &nbsp;
>
> ### C1. Regarding novelty
>
> **A1.** Thank you for pointing this out. First, we would like to clarify that we **do not claim** novelty in our usage of depth for NeRF nor warping-based consistency modeling, for we were very much aware of the previous methods [1][2] that you have mentioned, and had included both works in the Related Work section of our submitted paper.
>
> Instead, we claim novelty in how we demonstrate that source images warped to **unknown, random viewpoints that do not have ground truth images** can be effectively used as pseudo-ground truths for the regularization of NeRF’s geometry and appearance in a **few-shot setting,** as well as additional contributions such as **feature-level consistency**, **occlusion handling** and **progressive training strategies** that stabilize otherwise unstable sparse view NeRF optimization.  We elaborate in detail on how our method differs from the methods you have mentioned:
>
> **Reliance on additional depth priors**
>
> A key difference between methods [1][2] and GeCoNeRF (our method) is that both [1] and [2] rely on **additional depth priors** for their few-shot optimization of NeRF, while ours does not. [1] and [2] leverage ground truth 3D point cloud extracted from the structure-from-motion (SfM) process for input images as depth priors for NeRF. Moreover, they do not conduct any form of consistency modeling or warping, which makes them quite orthogonal to our method.
>
> For our model, we emphasize that our method does not rely on any external ground truth depth information or priors, and uses only sparse posed images for optimization, like the original NeRF. The depth we use for warping is the depth that has been recovered by NeRF under a few-shot setting.
>
> In summary, [1] and [2] give the model extra depth information: ours does not, but utilizes the depth already recovered by NeRF for warping. This is a major difference, and it gives our method far greater applicability and generalizability, as obtaining GT depth prior such as 3D point cloud is not trivial.
>
> **Warping to unseen viewpoints**
>
> We respectfully disagree with the reviewer's comment regarding how [3] and [4] warp source images to unseen views, as these methods warp source images **other known viewpoints** that have their own ground truth, not unseen viewpoints (random views without ground truth) in a sense that we use them in our paper. We find that when it comes to few-shot NeRF settings, standard consistency modeling between known views suffers a large drop in performance, due to scarcity of input viewpoints and increased difficulty in the warping procedure owing to wide viewpoint differences, heavy self-occlusions, and specular effects. We have added an experimental demonstration of this in Section 5.4 of our revised paper.
>
> Instead, in our model, source images are warped to truly **unobserved viewpoints** that do not have ground truths. Here we leverage a key advantage of NeRF, that depth can be rendered from any unobserved viewpoint - which in turn can be used to generate, with a carefully designed warping procedure (which we introduce in our paper), full RGB pseudo ground truth images in novel viewpoints. The gradient only flows to the **rendered color image** so that it becomes consistent with the warped patch. Therefore, our consistency modeling also has the additional advantage of regularizing **both color and geometry** of the given scene in novel views, unlike MVS’s consistency modeling which only refines geometry.
>
> Therefore, we emphasize that the novelty of our work comes from demonstrating how this effectively propagates detailed information from source images to nearby random, unobserved viewpoints. We show that it serves as a powerful regularization for otherwise unstable and divergent optimization of few-shot NeRF.
>
> **Experimental demonstration**
>
> To further reflect your comment, we have added a new Section 5.4, named **“Comparison to Consistency Modeling between Known Views”** in our revised paper, where we demonstrate how directly using source-to-source warping with consistency modeling (in a manner similar to standard MVS methods [3][4]) in a few-shot setting harms training and causes NeRF to act in a divergent manner.
>
> &nbsp;
>
> We hope our additional elaboration and experiment efforts could resolve your concerns regarding the novelty of our method. If you do not find them adequately convincing, please let us know and we will get back to you as soon as possible.

---

> ### Author Response · Authors · 2022-11-18
> **Follow up**
>
> Dear Reviewer 8ZEs,
>
> Thank you for your time and effort in reading our response! We hope our response has addressed your concerns. If you still feel unclear or concerned, please kindly let us know and we are more than glad to further clarify and discuss any further concerns. If you feel your concerns have been addressed, please kindly consider if it is possible to update your score.
>
> Thank you!
>
> Paper765 Authors

---

### Official Review · Reviewer_m8Jd · 2022-10-25

**Confidence:** 5
**Clarity, Quality, Novelty And Reproducibility:** See above
**Correctness:** 3
**Technical Novelty And Significance:** 2
**Empirical Novelty And Significance:** 2
**Recommendation:** 5

**Strength And Weaknesses:**

Pros:

1. The overall idea is tidy and easy to follow

2. Experiment results seem promising

Cons:

1. Some important references which address the multi-view photo-consistency are missing, e.g., MVSDF, NeuralWarp, and GeuNeuS. Specifically, MVSDF also applies feature consistency to constrain the neural rendering optimization.

2. Following Cons 1, I would expect a comparison between the proposed method with the explicit multi-view consistency formulation in MVSDF/NeuralWarp/GeuNeuS. (maybe an ablation study plus some explanations). From my perspective, they are all some kind of similar photoconsistency but implemented in different ways.

3. There are several wrong citations in the [Datasets and metrics] section.

**Summary Of The Paper:**

This paper introduces a geometric regularization technique for nerf training under a sparse view setting. The regularization is based on cross-view warping between seen/unseen views using rendered depth, where the warped image is utilized for supervision in feature space.



**Summary Of The Review:**

See above. I would like to hear the author's feedback on the missing references and I would raise my rating if my concerns could be well-addressed.

---

> ### Author Response · Authors · 2022-11-16
> **Official Comment to Reviewer m8Jd (2/2)**
>
> &nbsp;
>
> ### C2. Comparison between GeCoNeRF and methods mentioned in Cons 1. (Continued)
>
> **Comparison with NeuralWarp & GeoNeuS**
>
> As you have noted, NeuralWarp [2] and GeoNeuS [3] are very similar to one another in their modeling of photometric consistency. They warp source images to other source viewpoints and encourage the warped images to be consistent with ground truth images. Because the warping is governed by underlying geometry, this implicitly refines the geometry of the generated scene.
>
> However, these methods operate under dense input view settings, where the pose differences between neighboring source viewpoints are relatively small. Due to this reason, they use two approximations to simplify the consistency modeling procedure:
>
> - Patch warping between viewpoints can be approximated as planar homography warping, assuming the nearby pixels of a point as having the same depth
> - Pixel-level photometric difference between the warped patch and ground truth patch is supposed to be minimal and must be minimized
>
> However, in sparse view settings, where pose differences between source viewpoints are far more drastic, both of these assumptions do not apply well. In sparse view scenarios:
>
> - **Problem 1**: Warping cannot be naively approximated as planar homography warping - even minor depth differences cause pixel level self-occlusions when the source and target viewpoints are far apart.
> - **Problem 2**: View-dependent color differences (specularities) between input images are more drastic, due to their distance from each other. In this case, naively enforcing photometric consistency may not be beneficial and rather harmful for training.
>
> Thus, the main contributions our paper are novel methods designed to make warping-based consistency modeling possible in few-shot settings. To overcome **problem 1**, we use pixel-by-pixel reprojection and bilinear interpolation, combined with novel pixel-level occlusion masking to generate inversely warped patches. To address **problem 2**, we introduce feature-level consistency to let the model learn multi-level perceptual similarity, preventing the model from overfitting to source images’ colors (and thus suppressing specular effects) in unseen viewpoints. Furthermore, we introduce following novelties to maximize the effectiveness of our consistency regularization:
>
> - Instead of modeling between source viewpoints, whose large pose differences cause difficulty for warping, our method progressively warps source images to nearby **unobserved (novel) viewpoints** and model consistency between the rendered patch and warped patch in the said viewpoints.
> - We also point out that in our method, unlike [2][3], it is not the warped image but the **rendered image that is optimized to be consistent** with the warped patch - we do not let gradient flow to the warped patch, letting it serve as the pseudo ground truth. This is to give the network guidance and regularization in viewpoints where there are no ground truths.
>     - In Figure 5 of our paper, we demonstrate how this has an effect of letting our model recover fine geometry that were not possible for previous few-shot NeRF models. Also, we emphasize this has an additional benefit of allowing both geometry and appearance to be improved upon by the loss, unlike MVS consistency loss used in [2][3] which is only capable of influencing geometry.
>
> **Ablation study / Experimental results for comparison**
>
> - In the newly added Section 5.4 **“Comparison to Consistency Modeling between Known Views”** of our revised paper, we demonstrate how directly using source-to-source warping with consistency modeling (like NerualWarp, GeoNeuS) in a few-shot setting harms training and causes NeRF to act in a divergent manner.
> - In the Ablation Study of our revised paper, with the subsection **“Feature-level loss vs. pixel-level loss*”*** in Section 5.3, we also demonstrate how perceptual loss is superior to pixel-level photometric loss when it comes to few-shot, sparse view setting in regards to reconstruction quality.
>
> &nbsp;
>
> ### C3. Wrong Citations
>
> **A3.** We are sincerely grateful for your careful reading of our work and for pointing out these mistakes. We have fixed all erroneous references in the revised version of our paper.
>
> &nbsp;
>
> We hope our responses have addressed the concerns of the reviewer and we are happy to answer any further questions.
>
> &nbsp;
> ### **References**
>
> [1] Chibane, Julian, et al. “Stereo Radiance Fields (SRF): Learning View Synthesis for Sparse Views of Novel Scenes.”, CVPR 2021
>
> [2] Darmon, François, et al. “Improving neural implicit surfaces geometry with patch warping.”, CVPR 2022
>
> [3] Fu, Qiancheng, et al. “Geo-Neus: Geometry-Consistent Neural Implicit Surfaces Learning for Multi-view Reconstruction”, NeurIPS 2022

---

> ### Author Response · Authors · 2022-11-16
> **Official Comment to Reviewer m8Jd (1/2)**
>
> **General reply:** Thank you for your constructive review and helpful suggestions! We give a detailed response to your questions and comments down below. If any of our responses do not adequately address your concerns, please let us know and we will get back to you as soon as possible.
>
> &nbsp;
>
> ### C1. Missing references.
>
> **A1.** We are thankful for your valuable suggestion. We have included all the works you have mentioned in the revised version of our paper, in a newly made section in Related Works named “**Self-supervised photometric consistency”**. In the section, we discuss the difference between said methods and our GeCoNeRF, which we explain with greater detail in the response to your next comment, C2.
>
> &nbsp;
>
> ### C2. Comparison between GeCoNeRF and methods mentioned in Cons 1.
>
> **A2.** Thank you for bringing this up. We do agree that there is some similarity between our work and the mentioned methods in the usage of photometric consistency. However, we point out that are **multiple key differences** between GeCoNeRF and said methods (which we elaborate on below), with the main difference being that while other works only model consistency amongst known ground truth views, our work, specializing in few-shot sparse view setting, introduces a novel way to model consistency at **unknown, random viewpoints** that do not have ground truth. We also support our claims with multiple additional experiments in the revised version of our paper (Section 5.3 and 5.4), so we would greatly appreciate your effort if you could review it.
>
> **Comparison with MVSDF**
>
> Despite similar naming, the phrase “feature consistency” is used in MVSDF and GeCoNeRF to describe two very different methods. Let us elaborate:
>
> - Feature consistency in MVSDF refers to a prior that an important characteristic of a surface point is that its perceptual description (semantic feature) is consistent regardless of viewing direction (except for the viewpoints from which the point is occluded). This indicates that finding a location whose semantic features are consistent across viewpoints could be equivalent to finding the surface location, which serves as a powerful prior for geometry reconstruction. To leverage this, they first encode all known view images into immutable feature maps, and the network opts to find a surface point where the features of its projected locations are most consistent with one another. In other words, they implicitly **model geometry optimization as a matching problem**, leveraging **feature consistency between ground truth viewpoints**. A similar formulation could be found in SRF [1], which uses very similar prior in generalized NeRF architecture. Another important point to note is that this paper addresses only **“dense” input view** setting, where dozens of posed images are given as ground truth, and the above consistency modeling method serves as a signal for precise refinement of already well-captured geometry.
>
> - In our work, feature consistency indicates **patchwise feature-level reconstruction loss** applied at **unobserved viewpoints** between a rendered patch and a warped patch. The warped patch serves as the pseudo ground truth for the rendered patch. Our emphasis is that this effectively propagates detailed information from source images to nearby random, unobserved viewpoints, which we demonstrate to be a powerful regularization for otherwise unstable and divergent optimization of **few-shot** NeRF.

---

> ### Author Response · Authors · 2022-11-18
> **Follow up**
>
> Dear Reviewer m8Jd,
>
> Thank you for your time and effort in reading our response! We hope our response has addressed your concerns. If you still feel unclear or concerned, please kindly let us know and we are more than glad to further clarify and discuss any further concerns. If you feel your concerns have been addressed, please kindly consider if it is possible to update your score.
>
> Thank you!
>
> Paper765 Authors

---

> ### Author Response · Authors · 2022-11-29
> **Follow up reminder**
>
> Dear Reviewer m8Jd,
>
> We appreciate your time and effort in reading our response and revision! If you still have further concerns or feel unclear, please kindly let us know and we are happy to further clarify and discuss. If you feel your concerns have been addressed, we would appreciate it if you might kindly consider updating the score. As the discussion deadline is approaching, we really look forward to your feedback.
>
> Thank you!
>
> Paper765 Authors

---

> ### Comment · Reviewer_m8Jd · 2022-11-29
> **In reply to the rebuttal**
>
> Sorry for my late reply. Firstly I would like to thank the authors for the detailed response. I think the authors answered most of my questions and I am glad that the authors will add a new section in the related work to talk about photometric consistency.
>
> On the other hand, I am not quite convinced by the "few-shot" story after reading the comments/rebuttals. From my perspective, from "multi-view" to "few-shot" is natural if proper regularizations are applied, unless you are saying that strong semantics are introduced in the optimization to help complete the shape of unseen views (e.g., DietNeRF).
>
> Generally, I think the novelty of the paper is not very significant (photometric consistency has been widely used in many works), but the manuscript is nicely written with adequate experiments. I won't argue if the paper is finally accepted.

---

> > ### Author Response · Authors · 2022-11-30
> > **Rebuttal to reviewer m8Jd's comment**
> >
> > Dear Reviewer m8Jd,
> >
> > Thank you for your kind response. We greatly appreciate your time and effort in reading our work and our response. However, reading your response, there are additional clarifications we wish to make regarding our work’s novelty, and we would be truly grateful if you could read this comment, our final rebuttal, and give it a thought.
> >
> > In your comment, you said that “multiview to few-shot is natural if proper regularizations are applied”. If we understood it correctly, we take this statement as stating that translating techniques used in MVS tasks (such as photometric consistency) to few-shot NeRF setting is **trivial**, if used with minor regularizations.
> >
> > However, we respectfully disagree. We would like to point out and emphasize that the greatest difficulty for few-shot NeRF is not its inability to complete unseen regions, but its high **instability** and **divergent behaviors** that occur in multiple modes, as emphasized in our paper and attested in various works [1,2].  Please refer to the **depth map results** of baseline few shot **mip-NeRF in Figure 6** of our revised work to see clear visualizations of such “divergent behaviors”. The depth map of *Room* scene shows a thick wall of heavy artifacts filling the space that is should be empty (cloudy artifacts), and the depth map of *Fortress* scene reveals that it fails to capture the depth difference within the table or the object (geometry breakdown). Such unstable and divergent behaviors occur very commonly for few-shot NeRF, and this has been proven in numerous previous works [1,2,3] as well.
> >
> > Therefore, we call to attention that the main focus of most works in few-shot NeRF is how to **constrain and regularize** few-shot NeRF for stabilization. Such stabilization alone brings large performance gains. For this reason, InfoNeRF [1] and RegNeRF [2] solely propose regularization techniques without introducing any additional semantics, yet they demonstrate performances that completely dwarf Diet-NeRF [3] by a large margin.
> >
> > In this light, we clarify that the **main novelty and importance** of our work comes from demonstrating that consistency modeling, when used with our design choices, has the capability to **effectively stabilize and regularize** NeRF in few-shot setting - something which no other work has done before. We respectfully argue that this contribution is **novel** and **non-trivial**. Other works that use photometric consistency [4,5,6] utilize it to **refine** surfaces that are already geometrically accurate due to pretraining [4] and dense input view setting [4,5,6]: in our work, we implement it in a modified way (as stated in our rebuttal) with the purpose of **constraining and regularizing** geometry that would otherwise break down completely.
> >
> > To show that our contribution of modifying the consistency modeling is not trivial, we have qualitatively demonstrated in Section 5.4 and **Figure 9** of our revised paper that the traditional method of modeling photometric consistency between known views (as used in [4,5,6]) **fails** to resolve the divergent behavior of few-shot NeRF, while our method achieves it successfully.
> >
> > Among many methods that regularize and stabilize few-shot NeRF, our work for the first time demonstrates how to apply consistency modeling to achieve such regularization effect, and it displays state-of-the-art results. We hope this comment could resolve some of your concerns regarding our work’s novelty. We would also be very grateful if you could check the revised version of our paper, which has been greatly improved thanks to your constructive comments and suggestions. We are very thankful for your time and effort in reviewing our paper and responding to our comments.
> >
> >
> > Paper 765 authors
> >
> > &nbsp;
> > ### **References**
> >
> > [1] Kim, Mijeong, et al. “InfoNeRF: Ray Entropy Minimization for Few-Shot Neural Volume Rendering”, CVPR 2022
> >
> > [2] Niemeyer, Michael, et al. “RegNeRF: Regularizing Neural Radiance Fields for View Synthesis from Sparse Inputs,” CVPR 2022
> >
> > [3]  Jain, Ajay, et al. “Putting nerf on a diet: Semantically consistent few-shot view synthesis”, ICCV 2021
> >
> > [4] Darmon, François, et al. “Improving neural implicit surfaces geometry with patch warping.”, CVPR 2022
> >
> > [5] Fu, Qiancheng, et al. “Geo-Neus: Geometry-Consistent Neural Implicit Surfaces Learning for Multi-view Reconstruction”, NeurIPS 2022
> >
> > [6] Zhang, Jingyang, et al. “Learning Signed Distance Field for Multi-view Surface Reconstruction” ICCV 2021

---

### Official Review · Reviewer_w2pC · 2022-10-30

**Confidence:** 4
**Clarity, Quality, Novelty And Reproducibility:** The paper is well written and address…
**Correctness:** 4
**Technical Novelty And Significance:** 4
**Empirical Novelty And Significance:** 4
**Recommendation:** 8

**Strength And Weaknesses:**

Strength:
* The paper addresses a very relevant area of NeRF application to real world where capturing a scene with many images maybe impossible due to scene motion or having a 100-camera multi-camera system to capture a scene is economically and practically prohibitive.
* The paper is well written and all aspects are well discussed.
* The results look good and show improvement over existing works.

Weakness:
* No major weakness except a few "?" in the paper which can be resolved.

**Summary Of The Paper:**

The paper proposes a method to enable only sparse image set as an input to a NeRF network and enable rendering of artifact free novel views which are typically not possible using current existing methods. Their main contribution is to warp given input images to novel camera locations given the current estimate of the depth at novel view and minimize the error between the warped image and the NeRF rendered image at that novel viewpoint. The warped image is assumed to be a ground truth at the novel view-point. Since warping across translated views can lead to occlusion effects, a best estimate of occlusion mask is also calculated and used to mask out the cost error regions. The loss function is designed to be minimizing feature maps at multiple scales instead of direct pixel intensity difference because the pixel intensities can be different in reality if the scene contains non-Lambertian surfaces. The results are show to be better than standard and existing approaches.

**Summary Of The Review:**

Please see Strengths above.

---

> ### Author Response · Authors · 2022-11-16
> **Official Comment to Reviewer w2pC**
>
> &nbsp;
>
> ### S1. General reply
> Thank you for your thorough, perfect summary of our paper and positive comments. We are greatly encouraged by your assessment of our paper as relevant and well-written, acknowledging that our paper successfully addresses a crucially important weakness of NeRF in its reliance on hundreds of images with known poses which requires a prohibitive multi-camera setting. In complete agreement with your statement, we also believe that enabling few-shot training for NeRF is an important milestone for its applicability to real-life scenes and that our work is a step in that direction. If you have any additional concerns, please let us know and we will get back to you as soon as possible!
>
> &nbsp;
>
> ### W1. Missing references in the paper.
>
> **A1.** We are sincerely grateful for your careful reading of our work and for pointing the mistakes out. We have fixed all erroneous references and mistakes in the revised version of our paper.
>
> &nbsp;
>
> We hope our responses have addressed the concerns of the reviewer and we are happy to answer any further questions.

---

### Public Comment · ~Xuan_Wang2 · 2022-11-06
**few-shot?**

Why is it few-shot, not training from sparse view？

---

> ### Author Response · Authors · 2022-11-16
> **Regarding the phrase "few-shot"**
>
> Thank you for your comment. We call our task few-shot because both "training from sparse view" and "few-shot" are terminologies that are interchangeably used within this field to indicate the same task, as seen in previous works [1][2].
>
> [1] Kim, Mijeong, et al. “InfoNeRF: Ray Entropy Minimization for Few-Shot Neural Volume Rendering”, CVPR 2022
>
> [2] Jain, Ajay, et al. “Putting nerf on a diet: Semantically consistent few-shot view synthesis”, ICCV 2021

---

### Author Response · Authors · 2022-11-16
**General Response**

We would like to first thank the reviewers for the helpful suggestions and constructive reviews. We are greatly encouraged by their assessment of our work as **relevant** (w2pC), **novel** (8ZEs), and **clearly demonstrating its contributions** (Copi), displaying **state-of-the-art** (Copi) **promising results** (w2pC, m8Jd, 8ZEs) with **extensive engineering efforts** (Copi), and their unanimous judgment of our paper as **clear and organized** (8ZEs), **well-written** (w2pC,  Copi), **easy to follow** (m8Jd) with **all aspects well-discussed** (w2pC). We are grateful that they saw significance in our quantitative and qualitative improvement over our baselines, achieving few-shot novel view synthesis quality competitive to current SOTA models. We carefully address each concern given by reviewers with detailed explanations and supporting experimental results.

Following the insightful comments given by the reviewers to improve our work, we have made improvements to the revised version of our paper. The writing has been modified according to the reviewers’ suggestions in order to clarify our motivations and contributions. We summarize the updates below.

- We have fixed all erroneous references, typos, and compiling errors.
- We have made improvements in Figures 1 and 2 to provide further clarification on our model pipeline.
- We have made minor adjustments to equations and notations for clarification.
- We have added an extended discussion in **“Few-shot NeRF”** of Section 2 (Related Work), as suggested by reviewer 8ZEs, regarding the limitations of previous methods and comparison to our method.
- We have replaced the subsection **“Monodepth estimation”** with **“Self-supervised photometric consistency”** in Section 2 (Related Work), as suggested by reviewer 8ZEs,  which references other contemporary methods that use consistency modeling along with brief comparisons to our work.
- We have added Figure 4 in Section 4.3 (Consistency Modeling) to provide an intuitive visualization of our occlusion mask generation procedure.
- We have added more qualitative results on the NeRF-synthetic dataset in Figure 5 of Section 5.2.
- We have added more qualitative results on the LLFF dataset in Figure 6 of Section 5.2.
- We have added further elaboration regarding comparison with previous SOTA methods in **“Qualitative comparisons.”** of Section 5.2.
- We have made multiple modifications in Section 5.3 (Ablation Study):
    - We have expanded the main ablation study table with more detailed information.
    - We have divided the section into subsections describing each ablative experiment.
    - We have added subsection **“Progressive training strategies”**, following the comment of reviewer Copi, along with quantitative ablation results, to demonstrate its effectiveness.
    - We have added Figure 8 to the subsection **“Feature-level loss vs. pixel-level loss.”** to qualitatively compare the effects of pixel-level photometric loss with feature-level consistency modeling.
- To address reviewers’ concerns regarding our novelty (m8Jd, 8ZEs) and comparison to existing MVS methods, we have added a new section named **“Comparison to Consistency Modeling between Known Views”** as Section 5.4 in our revision to provide experimental results and discussion regarding usage of standard MVS consistency modeling loss, which models consistency between known views, in few-shot NeRF setting.

We would be grateful if you could review the renewed version of our paper.  If there are any comments that we did not adequately address despite the revision, they will be thoroughly reflected in the final version of our paper. Thank you for your helpful and constructive reviews.

---

### Decision · Program_Chairs · 2023-01-20

**Decision:**

Reject

**Justification For Why Not Higher Score:**

The area chair has read the paper, reviews and rebuttal and believes that the contribution of the present paper may be too minor  for the ICLR audience. The proposed NeRF model does not learn representations that transfer across scenes in any way. We believe the contributions of this work more enthusiastically accepted by  the graphics community because the paper has not shown how to contribute to  visual recognition or representation learning that are  related themes of ICLR.




**Justification For Why Not Lower Score:**

N/A

**Metareview: Summary, Strengths And Weaknesses:**

The paper proposes a method to enable a NeRF model to use a sparse set of images of a specific scene and render images from novel viewpoints of that scene. Their main contribution of the paper is to warp given input images to novel camera locations given estimated  depth and minimize the error between the warped image and the NeRF rendered image at that novel viewpoint. Since warping across views can lead to occlusions, a best estimate of occlusion mask is also calculated and used to mask out the loss at the estimated occluded regions. The loss function is designed to be minimizing feature maps at multiple scales instead of direct pixel intensity difference. The results show a boost in the quality of rendered views over previous models.
The reviewers overall agree on the contribution of the paper and its improved quality of rendered images. One reviewer misunderstood the contribution thinking that the novel  viewpoints are given, but the authors clearly clarify this misconception in their rebuttal. They point out some missing or wrong citations that the rebuttal of the authors successfully addressed. The area chair has read the paper, reviews and rebuttal and believes that the contribution of the present paper may be too minor  for the ICLR audience. The proposed NeRF model does not learn representations that transfer across scenes in any way. We believe the contributions of this work more enthusiastically accepted by  the graphics community because the paper has not shown how to contribute to  visual recognition or representation learning that are  related themes of ICLR. The authors are encouraged to submit their work in upcoming relevant venues where papers that address single scene NeRF fitting are presented, such as SIGGRAPH or ICCV.


**Summary Of Ac-Reviewer Meeting:**

N/A